# Phase Separation Phenomena in Lightly Cu-Doped A-Site-Ordered Quadruple Perovskite NdMn_7_O_12_

**DOI:** 10.3390/molecules30234561

**Published:** 2025-11-26

**Authors:** Alexei A. Belik, Ran Liu, Kazunari Yamaura

**Affiliations:** 1Research Center for Materials Nanoarchitectonics (MANA), National Institute for Materials Science (NIMS), Namiki 1-1, Tsukuba 305-0044, Ibaraki, Japan; liu.ran@sanken.osaka-u.ac.jp (R.L.); yamaura.kazunari@nims.go.jp (K.Y.); 2Graduate School of Chemical Sciences and Engineering, Hokkaido University, North 10 West 8, Kita-ku, Sapporo 060-0810, Hokkaido, Japan; 3Institute of Scientific and Industrial Research, Osaka University, Mihogaoka 8-1, Ibaraki 567-0047, Osaka, Japan

**Keywords:** A-site-ordered quadruple perovskites, orbital order, phase separation

## Abstract

A-site-ordered quadruple perovskite manganites, AMn_7_O_12_, show many interesting physical phenomena, including orbital and spin modulations, spin-induced multiferroic properties, and competitions between different magnetic ground states. Doping with Cu^2+^ can result in colossal magnetoresistance properties, ferrimagnetism, and additional structural modulations producing electric–dipole helicoidal textures. Many previous works have focused on large-concentration doping, reaching ACu_3_Mn_4_O_12_ compositions. Small-concentration doping has been investigated in a limited number of systems, e.g., in BiCu*_x_*Mn_7−*x*_O_12_. In this work, we investigated solid solutions of NdCu*_x_*Mn_7−*x*_O_12_ with *x* = 0.1, 0.2, and 0.3, prepared at 6 GPa and 1500 K. Specific heat measurements detected three magnetic transitions at *x* = 0 (at *T*_N3_ = 9 K, *T*_N2_ = 12 K, and *T*_N1_ = 84 K) and two transitions at *x* = 0.1 (at *T*_N2_ = 10 K and *T*_N1_ = 78 K), while only one transition was found at *x* = 0.2 (*T*_N1_ = 72 K) and *x* = 0.3 (*T*_N1_ = 65 K). Differential scanning calorimetry (DSC) measurements showed sharp and strong peaks near *T*_OO_ = 664 K at *x* = 0, corresponding to an orbital-order (OO) structural transition from *I*2/*m* to *Im*-3 symmetry. DSC anomalies were significantly broadened and their intensities were significantly reduced at *x* = 0.1–0.3, and structural transitions were observed near *T*_OO_ = 630 K at *x* = 0.1, *T*_OO_ = 600 K at *x* = 0.2, and *T*_OO_ = 570 K at *x* = 0.3. The *x* = 0.1 sample clearly showed double-peak features on the DSC curves near *T*_OO_ because of the presence of two close phases. High-resolution synchrotron powder X-ray diffraction studies gave strong evidence that phase separation phenomena took place in the *x* = 0.1–0.3 samples, where two *I*2/*m* phases with an approximate ratio of 1:1 were present (e.g., *a* = 7.47143 Å, *b* = 7.36828 Å, *c* = 7.46210 Å, and *β* = 90.9929° for one phase and *a* = 7.46596 Å, *b* = 7.37257 Å, *c* = 7.45756 Å, and *β* = 90.9328° for the second phase at *x* = 0.3). The Curie–Weiss temperature changed from negative (for *x* = 0, 0.1, and 0.2) to positive (for *x* = 0.3). *T*_OO_, *T*_N1_, the Curie–Weiss temperature, and magnetization (at 5 K and 70 kOe) changed almost linearly with *x*.

## 1. Introduction

Classical perovskite-structure materials with the stoichiometry of ABO_3_ have large A cations and smaller B cations [1,2,3,4,5]. The stability of the perovskite structure is determined by the relative sizes of A and B cations and is usually understood in terms of the Goldschmidt tolerance factor [2]. The perovskite structure (at least, in the oxide form) is quite flexible and can adapt (and form ordered structures) to situations where some A cations (defined as A′ and A″) have similar sizes to B cations. Such adaptations take place through the *a*^+^*a*^+^*a*^+^ tilt (in Glazer’s notation [6]) in the case of A-site-ordered quadruple perovskites, AA′_3_B_4_O_12_ [7,8,9,10,11,12,13], and through the *a*^+^*a*^+^*c*^−^ tilt in the case of A-site columnar-ordered quadruple perovskites, A_2_A′A″B_4_O_12_ [13]. Strong octahedral tilting produces square planar coordination (as the first coordination sphere) around the A′ sites in both subfamilies.

If A′ = B = Mn in A-site-ordered quadruple AA′_3_B_4_O_12_ perovskites, an interesting class of perovskite manganites is formed, AMn_3_Mn_4_O_12_ or AMn_7_O_12_ in short form [12]. Depending on the oxidation state of the A cation, Mn cations can take different oxidation states (at the B sites) from +3.5 for A^+^ [14,15] to +3.25 for A^2+^ [16] to +3 for A^3+^ [17,18,19,20,21,22,23,24,25,26,27,28,29,30,31,32,33,34], while Mn cations usually take the +3 oxidation state at the A′ site. The average oxidation state of Mn at the B sites determines the structural and magnetic behaviors of AMn_7_O_12_. All AMn_7_O_12_ compounds crystallize in the parent structure of the AA′_3_B_4_O_12_ quadruple perovskites with space group *Im*-3 at high temperatures. However, with decreasing temperature, different structural distortions take place. For example, all members of the trivalent subfamily A^3+^Mn_7_O_12_ (except BiMn_7_O_12_ [20,21,22]) crystallize in the monoclinic space group *I*2/*m* at room temperature (RT). The *Im*-3 to *I*2/*m* transition is driven by the orbital ordering (OO) of Jahn–Teller active Mn^3+^ cations at the B sites.

The average oxidation state of Mn at the B sites can be continuously changed in different solid solutions, such as A^+^Mn_7_O_12_–A^2+^Mn_7_O_12_ and A^2+^Mn_7_O_12_–A^3+^Mn_7_O_12_ [35,36,37]. The average oxidation state of Mn at the B sites can also be controlled through Cu^2+^ doping, producing ACu*_x_*Mn_7−*x*_O_12_ solid solutions. Cu^2+^ cations are usually selected as dopants because Cu^2+^ cations occupy square planar sites similar to Mn^3+^ cations and do not disturb/dilute Mn in the B sublattice [38,39,40,41,42,43,44,45,46,47,48,49,50,51,52,53,54,55]. With Cu^2+^ doping, the Mn^3+^/Mn^4+^ ratio is only changed in the B sublattice. Large Cu^2+^ doping, e.g., reaching ACu_3_Mn_4_O_12_ [46,47,48,49,50,51,52,53,54,55], produces ferrimagnetic (FiM) properties with high magnetic transition temperatures above RT. Such A^2+^Mn_7_O_12_ and A^3+^Mn_7_O_12_ manganites also show good catalytic properties [10].

It was recently found that light Cu^2+^ doping of BiMn_7_O_12_ can introduce electric-dipole helicoidal textures [56], mesoscopic helices of polar domains [57], and complex temperature–composition phase diagrams [58,59]. Light Cu^2+^ doping of other members of the trivalent subfamily A^3+^Mn_7_O_12_ has been poorly investigated. LaCu*_x_*Mn_7−*x*_O_12_ solid solutions were only studied at small doping levels [60]. However, the structural evolution at RT was only reported as a function of *x* without detailed physical properties [60]. It was found that the monoclinic *I*2/*m* structure of the parent LaMn_7_O_12_ was realized at *x* = 0.12–0.32, the *R*-3 structure, observed in Ca^2+^Mn_7_O_12_ [16], was realized at *x* = 0.52–0.84, and the cubic *Im*-3 parent structure of AA′_3_B_4_O_12_ quadruple perovskites was realized at *x* = 1–3 [60]. Detailed effects of the low doping levels of Cu^2+^ were mainly investigated in CaMn_7_O_12_ [61,62,63,64,65], where incommensurate crystal and magnetic structures are highly sensitive to such doping. We emphasize that previous structural studies with neutron diffraction [44,51,53,56,58,59,61] showed that Cu^2+^ cations are always localized at the square planar A′ site due to the strong Jahn–Teller effect of Cu^2+^.

Therefore, in this work, we prepared and investigated NdCu*_x_*Mn_7−*x*_O_12_ solid solutions at small doping levels of *x* = 0.1, 0.2, and 0.3. We found that the NdCu*_x_*Mn_7−*x*_O_12_ solid solutions preserve the monoclinic structure of the parent NdMn_7_O_12_. However, the phase separation phenomenon was observed, where two close monoclinic *I*2/*m* phases were present in NdCu*_x_*Mn_7−*x*_O_12_ solid solutions. The structural divergence between the two phases monotonically increases with increasing *x*. The Curie–Weiss temperature changes from negative (for *x* = 0, 0.1, and 0.2) to positive (for *x* = 0.3). The structural transition temperature (*T*_str_ = *T*_OO_), the first magnetic transition temperature (*T*_N1_), the Curie–Weiss temperature, and magnetization (at 5 K and 70 kOe) change almost linearly with *x*.

## 2. Results and Discussion

### 2.1. Magnetic Properties

The *χ* versus *T* curves of NdMn_7_O_12_ clearly showed three magnetic anomalies (Figure 1) at *T*_N3_ = 9 K, *T*_N2_ = 12 K, and *T*_N1_ = 84 K [33,34]. A transition at *T*_N1_ = 84 K shows sharp increases in the *χ* values (especially at *H* = 100 Oe). Then, there are additional small increases in the *χ* values at *T*_N2_ = 12 K and *H* = 10 kOe and small drops in the *χ* values at *T*_N3_ = 9 K and *H* = 10 kOe. The behavior of the *χ* versus *T* curves of NdMn_7_O_12_ at *H* = 10 kOe agrees well with magnetic structures found by neutron diffraction studies [33]. Below *T*_N1_, a FiM transition takes place on the Mn sublattice at the A′ site (with one Mn^3+^ uncompensated moment per unit cell), while an antiferromagnetic (AFM) order is realized on the Mn sublattice at the B site. The presence of one uncompensated site leads to a strong rise in *χ* values below *T*_N1_. Below *T*_N2_, small spin canting of the AFM spins and incommensurate spin modulation appear at the B site—this leads to an additional small increase in *χ* values below *T*_N2_. Below *T*_N3_, the Nd^3+^ sublattice is ordered with moments being antiparallel to the FiM moments of Mn^3+^ at the A′ site—this leads to small drops in *χ* values at *T*_N3_ = 9 K.

The *χ* versus *T* curves of NdCu_0.1_Mn_6.9_O_12_ (Figure 2) showed the first strong increases in the *χ* values below *T*_N1_ = 78 K. However, the additional upturn (observed near 12 K in NdMn_7_O_12_) nearly disappeared, while the small drops survived (near 10 K). Therefore, we can suggest that an incommensurate ordering (observed below *T*_N2_ = 12 K in NdMn_7_O_12_) is suppressed, and only magnetic transitions, corresponding to *T*_N1_ and *T*_N3_ in NdMn_7_O_12_, survived. The Nd sublattice remained undisturbed; therefore, the Nd sublattice can behave similarly in NdMn_7_O_12_ and NdCu_0.1_Mn_6.9_O_12_. Cu^2+^ cations should be located at the A′ sites; however, Cu^2+^ doping introduces Mn^4+^ cations at the B sites of NdCu_0.1_Mn_6.9_O_12_. Therefore, the incommensurate ordering and spin canting at the B sites could be strongly affected by Cu^2+^ doping. In comparison with NdMn_7_O_12_, *T*_N2_ disappears in NdCu_0.1_Mn_6.9_O_12_. However, we use sequential numbering of magnetic phase transitions with *T*_N2_ = 10 K and *T*_N1_ = 78 K for NdCu_0.1_Mn_6.9_O_12_.

The low-temperature drops are further suppressed in NdCu_0.2_Mn_6.8_O_12_ (Figure 3) and completely disappear in NdCu_0.3_Mn_6.7_O_12_ in all magnetic fields (Figure 4). The transition temperatures are summarized in Table 1, and the first transition temperature (*T*_N1_) almost linearly decreases with increasing *x* in the NdCu*_x_*Mn_7−*x*_O_12_ solid solutions (Figure 5a).

At high temperatures, the inverse magnetic susceptibilities followed the Curie–Weiss law. To extract the Curie–Weiss parameters, we fitted the field-cooled inverse magnetic susceptibilities (at *H* = 10 kOe) between 200 K and 350 K. The fitting parameters are summarized in Table 1 [66]. The Curie–Weiss temperature changes from negative (for *x* = 0, 0.1, and 0.2) to positive (for *x* = 0.3), and it follows a nearly linear change with *x* (Figure 5b). The increase in Cu^2+^ doping increases the concentration of Mn^4+^ at the B sites. Therefore, the concentration of ferromagnetic (FM) interactions through the double-exchange mechanism between Mn^3+^ and Mn^4+^ cations [67] also increases. This is reflected in the changes in the Curie–Weiss temperature.

Isothermal magnetization curves (*M* versus *H*) at *T* = 5 K are given on Figure 6. These are typical for ferrimagnets with well-defined hysteresis near the origin and gradual continuous increases in magnetization at higher magnetic fields due to the AFM B sublattice. There were monotonic increases in the remnant magnetization and coercive fields with *x* and a nearly linear increase in the magnetization values, *M*_S_ (at *T* = 5 K and *H* = 70 kOe), with *x* (Figure 5b). Some parameters of the *M* versus *H* curves are summarized in Table 1.

The results of the specific heat measurements of NdCu*_x_*Mn_7−_*_x_*O_12_ with *x* = 0, 0.1, 0.2, and 0.3 at different magnetic fields are shown in Figure 7. In agreement with the *χ* versus *T* curves, NdMn_7_O_12_ showed two sharp anomalies at *T*_N2_ = 12 K and *T*_N1_ = 84 K and a shoulder-like anomaly at *T*_N2_ = 9 K (the inset of Figure 7a). The double-peak anomalies near 10–15 K survived in NdMn_7_O_12_ at *H* = 90 kOe. The specific heat anomaly near *T*_N2_ = 10 K was significantly suppressed in NdCu_0.1_Mn_6.9_O_12_ (Figure 7b) and was completely suppressed in NdCu_0.2_Mn_6.8_O_12_ (Figure 7c) and NdCu_0.3_Mn_6.7_O_12_ (Figure 7d). On the other hand, the magnetic entropy that was released near *T*_N2_ in NdMn_7_O_12_ and NdCu_0.1_Mn_6.9_O_12_ moved to higher temperatures (up to about 45 K) in NdCu_0.2_Mn_6.8_O_12_ and NdCu_0.3_Mn_6.7_O_12_ (the inset of Figure 7d). The *C*_p_/*T* values of NdCu_0.2_Mn_6.8_O_12_ and NdCu_0.3_Mn_6.7_O_12_ were nearly the same between 2 K and 60 K.

### 2.2. High-Temperature Structural Transitions

The high-temperature behavior of NdCu*_x_*Mn_7−_*_x_*O_12_ with *x* = 0, 0.1, 0.2, and 0.3 was investigated with differential scanning calorimetry (DSC). NdMn_7_O_12_ showed very sharp and strong DSC anomalies near a structural (str) phase transition with *T*_str_ = 684 K (defined from peak positions on heating curves) [34]. The phase transition temperature remained nearly the same during cycling (three runs) (Figure 8a,b). This structural transition corresponds to the symmetry change from *I*2/*m* (below *T*_str_) to *Im*-3 (above *T*_str_) and is related to the orbital order (OO) of Mn^3+^ cations at the B sites below *T*_str_ [12,34].

DSC anomalies were already significantly broadened in NdCu_0.1_Mn_6.9_O_12_ (Figure 8c,d), and their intensities were strongly suppressed in comparison with NdMn_7_O_12_. In addition, the first heating DSC curve showed three peaks, while the second and third heating DSC curves showed two peaks. The cooling curves were reproducible and showed two peaks. The appearance of two peaks could be explained by the phase separation discussed in the next part. The difference between the first and subsequent DSC heating curves suggests the presence of “annealing” effects, when metastable states, obtained through quenching at a high pressure of 6 GPa, transform to more stable states. The behavior shown in the first and subsequent heating DSC curves was observed, for example, in BiMn_7_O_12_ [22].

DSC anomalies became even broader in NdCu_0.2_Mn_6.8_O_12_ (Figure 9a,b). There was irreproducibility between the first and subsequent heating curves, while the cooling DSC curves were almost reproducible. In the case of NdCu_0.3_Mn_6.7_O_12_, no clear DSC anomalies were observed in the heating curves (Figure 9c), while the cooling curves enabled very broad anomalies starting from about 570 K to be detected (Figure 9d). The structural phase transition temperatures decreased almost linearly with *x* in the NdCu*_x_*Mn_7−_*_x_*O_12_ solid solutions (Figure 5a).

### 2.3. Structural Properties

The NdCu*_x_*Mn_7−_*_x_*O_12_ samples with *x* = 0, 0.1, 0.2, and 0.3 did not contain any impurity phases because no impurity peaks could be seen, even for high-resolution, high-intensity synchrotron powder X-ray diffraction (PXRD) data, indicating the high quality of the samples. Reflections on laboratory PXRD data could be readily explained/indexed by assuming the presence of one monoclinic phase with the *I*2/*m* symmetry similar to undoped NdMn_7_O_12_ [33,34]. However, attempts to fit the synchrotron PXRD data with one *I*2/*m* phase were not successful, as one phase could not explain all reflection splitting (see the inset of Figure 10). All attempts to reduce the symmetry (for example, to triclinic symmetry) in a one-phase model were also unsuccessful. On the other hand, all reflection splitting in all samples could be well explained by assuming the presence of two *I*2/*m* phases with slightly different lattice parameters (Appendix A and Table 2). The DSC results for the *x* = 0.1 sample (Figure 8c,d) support the conclusion about the presence of two close phases, as each phase can show slightly different *T*_str_ values. The intensities of the two DSC peaks in the *x* = 0.1 sample (on the cooling curves and on the second and third heating curves) were comparable with each other, in agreement with the approximate 1:1 ratio of the two monoclinic phases found through synchrotron PXRD. Therefore, in general, the DSC results can provide valuable information about phase separation in the absence of high-resolution synchrotron XRPD data. However, in the *x* = 0.2 and 0.3 samples, the DSC anomalies became so broad that different *T*_str_ values were not resolvable.

Figure 11a shows the compositional dependence of the monoclinic lattice parameters of NdCu*_x_*Mn_7−_*_x_*O_12_ with *x* = 0.1, 0.2, and 0.3. In all cases, the second *I*2/*m* phase had a smaller monoclinic distortion in comparison with the first *I*2/*m* phase in the sense that the monoclinic *β* angle (of the second *I*2/*m* phase) was closer to 90°, and the difference between the lattice parameters (*a*_M_ − *b*_M_ and *c*_M_ − *b*_M_) was smaller for the second *I*2/*m* phase. In addition, systematic increases in the splitting of the lattice parameters between the second and first *I*2/*m* phases were observed with an increasing *x* value (Figure 11b). Such monotonic changes in the divergence between the two monoclinic phases provide indirect support that the observed phase separation is an intrinsic property of the NdCu*_x_*Mn_7−_*_x_*O_12_ system in the compositional range of *x* = 0.1–0.3. We found that in the compositional range of *x* = 0.5–0.8, the NdCu*_x_*Mn_7−_*_x_*O_12_ solid solutions (prepared in the same conditions as the *x* = 0.1–0.3 samples) had the *R*-3 symmetry and showed no evidence of phase separation (including the DSC results where only one DSC peak was found and high-resolution synchrotron PXRD). The NdCu*_x_*Mn_7−_*_x_*O_12_ solid solutions crystallized in the cubic *Im*-3 structure for *x* = 1.1–3.0, and we did not see any evidence of phase separation in the high-resolution synchrotron PXRD; no phase separation was also found in the literature in this compositional range [50,51,54]. Therefore, the phase separation phenomenon depends on the chemical composition/symmetry and is not an artifact of the preparation conditions, mixing procedures, or other factors. The real reason for the phase separation in the 0.1–0.3 samples is not clear and will require additional studies in the future. Magnetic properties were not affected by phase separation, and magnetic measurements could not provide any evidence of phase separation, likely because the magnetic transition temperatures of the two monoclinic phases were too close to each other and all magnetic and specific heat anomalies from the two phases severely overlapped. Monoclinic phases with different lattice parameters were observed in the A^3+^Mn_7_O_12_ systems when the oxygen content was varied (for A = Pr [28]) or small amounts of Mn^4+^ cations were introduced through A-site deficiency (for A = Bi [23,24]). Therefore, one of the driving forces for phase separation in the NdCu*_x_*Mn_7−_*_x_*O_12_ solid solutions could be small variations in the oxygen content of the two monoclinic phases. However, the Mn^3+^/Mn^4+^ distribution and ratio (which depend on the oxygen content and Cu^2+^ doping levels) or lattice strain could also contribute to the phase separation. We also note that phase separation was reported in cubic CdCu_3_Mn_4_O_12_ [45], prepared under high-pressure conditions; however, in this case, the formation of a CuO impurity (and compositional shifts) could have contributed to the effect. Phase separation was also observed in some of the cubic CaCu_3_Ti_4_O_12_ samples [68], prepared at ambient pressure.

## 3. Materials and Methods

NdCu*_x_*Mn_7−*x*_O_12_ samples with *x* = 0, 0.1, 0.2, and 0.3 were prepared from stoichiometric mixtures of Nd_2_O_3_ (Rare Metallic Co., Tokyo, Japan, 99.9%), CuO (Rare Metallic Co., Tokyo, Japan, 99.9%), MnO_2_ (Alfa Aesar, Ward Hill, MA, USA, 99.99%), and Mn_2_O_3_. Single-phase Mn_2_O_3_ was prepared from a commercial MnO_2_ chemical (Rare Metallic Co., Tokyo, Japan, 99.99%) by annealing in air at 923 K for 24 h. The synthesis was performed at 6 GPa and 1500 K for 2 h in sealed Au capsules using a belt-type, high-pressure HP instrument. After annealing at 1500 K, the samples were cooled down to room temperature (RT) by turning off the heating current, and the pressure was slowly released.

Powder X-ray diffraction (XRPD) data were collected at RT on a MiniFlex600 diffractometer (Rigaku, Tokyo, Japan) using CuKα radiation (a 2*θ* range of 5–100°, a step width of 0.02°, and a scan speed of 2 °/min). RT synchrotron XRPD data were measured on the BL15XU beamline (the former NIMS beamline) of SPring-8 [69] between 3.04° and 59.33° at 0.003° intervals in 2*θ* with a wavelength of *λ* = 0.65298 Å. The samples were placed into open Lindemann glass capillary tubes (inner diameter: 0.1 mm), which were rotated during the measurements. The Rietveld analysis of all XRPD data and indexing were performed using the RIETAN-2000 program [70].

Magnetic measurements were performed on an SQUID magnetometer (Quantum Design MPMS-XL-7T, San Diego, CA, USA) between 2 K and 350 K in applied fields of 100 Oe and 10 kOe under both zero-field-cooled (ZFC) and field-cooled on cooling (FCC) conditions. Magnetic field dependence was measured at *T* = 5 K between −70 and 70 kOe. The base temperature of the MPMS-XL-7T magnetometer was 10 K, meaning that samples were inserted slowly into 10 K, and samples moved through a magnet (with finite trapped magnetic fields) below their ordering temperatures. This procedure could be the reason for negative initial magnetization in some ZFC curves (at *H* = 100 Oe), even though the magnet reset procedure was applied before each ZFC measurement.

Specific heat, *C*_p_, was measured during cooling from 300 K to 2 K at a magnetic field of zero and from 150 K to 2 K at different magnetic fields through a pulse relaxation method using a commercial calorimeter (Quantum Design PPMS, San Diego, CA, USA). All magnetic and specific heat measurements were performed using pieces of pellets.

Differential scanning calorimetry (DSC) curves of powder samples were recorded on a Mettler Toledo DSC1 STAR^e^ system (Columbus, OH, USA) between 297 K and 703 K in open Al capsules with a heating/cooling rate of 10 K/min. Three DSC runs were performed to check the reproducibility.

## 4. Conclusions

Solid solutions of NdCu*_x_*Mn_7−*x*_O_12_ with *x* = 0, 0.1, 0.2, and 0.3 were synthesized by a high-pressure, high-temperature method at 6 GPa and 1500 K. The specific heat and magnetic measurements uncovered three magnetic transitions at *x* = 0 (at *T*_N3_ = 9 K, *T*_N2_ = 12 K, and *T*_N1_ = 84 K), two transitions at *x* = 0.1 (at *T*_N2_ = 10 K and *T*_N1_ = 78 K), and only one transition at *x* = 0.2 (at *T*_N1_ = 72 K) and *x* = 0.3 (at *T*_N1_ = 65 K). The DSC measurements showed sharp and strong peaks near *T*_OO_ = 664 K at *x* = 0, while the DSC anomalies were significantly broadened and their intensities were significantly reduced in other samples. Structural transitions were detected near *T*_OO_ = 630 K at *x* = 0.1, *T*_OO_ = 600 K at *x* = 0.2, and *T*_OO_ = 570 K at *x* = 0.3. The first magnetic transition temperature, the Curie–Weiss temperature, the structural transition temperature, and magnetization (at 5 K and 70 kOe) changed almost linearly with *x*. In addition, the Curie–Weiss temperature changed from negative (for *x* = 0, 0.1, and 0.2) to positive (for *x* = 0.3). The high-resolution synchrotron powder X-ray diffraction and DSC studies provided strong evidence that a phase separation phenomenon took place in the *x* = 0.1–0.3 samples, where two *I*2/*m* phases with an approximate ratio of 1:1 were present.

## Figures and Tables

**Figure 1 molecules-30-04561-f001:**
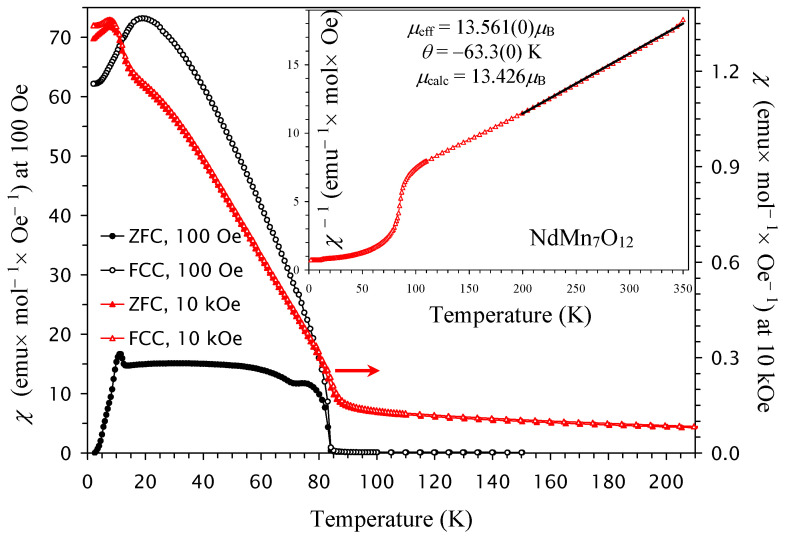
ZFC (filled symbols) and FCC (empty symbols) dc magnetic susceptibility curves (*χ* = *M*/*H*) of NdMn_7_O_12_ measured at *H* = 100 Oe (black symbols; the left-hand axis) and *H* = 10 kOe (red symbols; the right-hand axis). The inset shows the 10 kOe FCC *χ*^−1^ versus *T* curve with the Curie–Weiss fit (black line) between 200 K and 350 K; the parameters of the fit (*μ*_eff_ and *θ*) are given in the figure.

**Figure 2 molecules-30-04561-f002:**
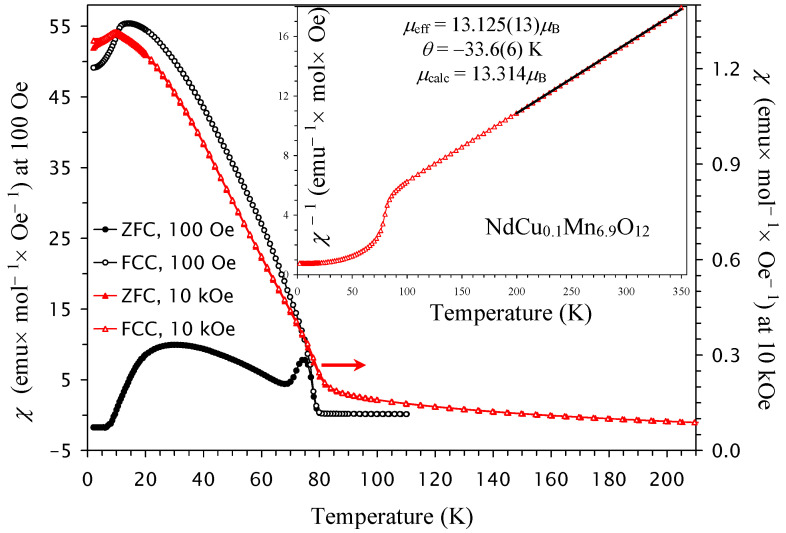
ZFC (filled symbols) and FCC (empty symbols) dc magnetic susceptibility curves (*χ* = *M*/*H*) of NdCu_0.1_Mn_6.9_O_12_ measured at *H* = 100 Oe (black symbols; the left-hand axis) and *H* = 10 kOe (red symbols; the right-hand axis). The inset shows the 10 kOe FCC *χ*^−1^ versus *T* curve with the Curie–Weiss fit (black line) between 200 K and 350 K; the parameters of the fit (*μ*_eff_ and *θ*) are given in the figure.

**Figure 3 molecules-30-04561-f003:**
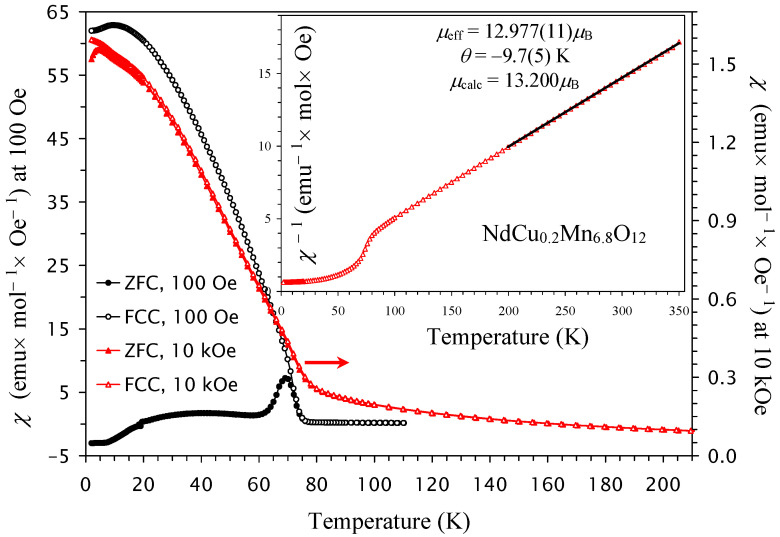
ZFC (filled symbols) and FCC (empty symbols) dc magnetic susceptibility curves (*χ* = *M*/*H*) of NdCu_0.2_Mn_6.8_O_12_ measured at *H* = 100 Oe (black symbols; the left-hand axis) and *H* = 10 kOe (red symbols; the right-hand axis). The inset shows the 10 kOe FCC *χ*^−1^ versus *T* curve with the Curie–Weiss fit (black line) between 200 K and 350 K; the parameters of the fit (*μ*_eff_ and *θ*) are given in the figure.

**Figure 4 molecules-30-04561-f004:**
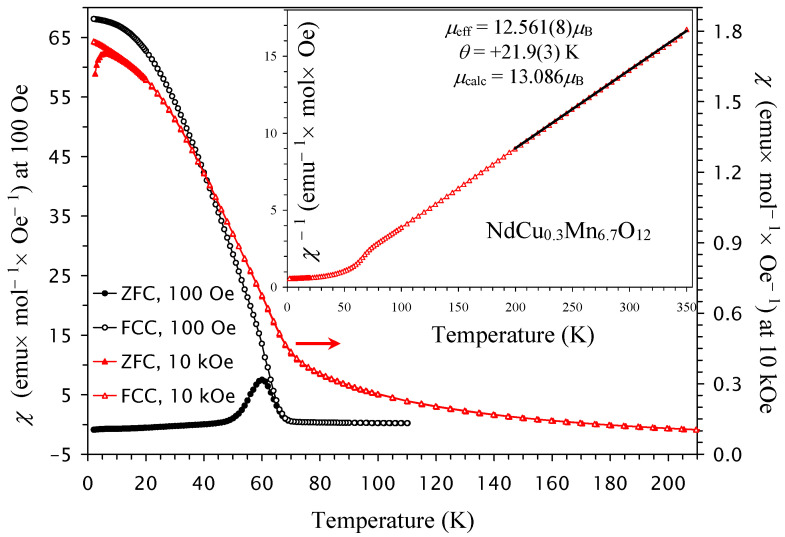
ZFC (filled symbols) and FCC (empty symbols) dc magnetic susceptibility curves (*χ* = *M*/*H*) of NdCu_0.3_Mn_6.7_O_12_ measured at *H* = 100 Oe (black symbols; the left-hand axis) and *H* = 10 kOe (red symbols; the right-hand axis). The inset shows the 10 kOe FCC *χ*^−1^ versus *T* curve with the Curie–Weiss fit (black line) between 200 K and 350 K; the parameters of the fit (*μ*_eff_ and *θ*) are given in the figure.

**Figure 5 molecules-30-04561-f005:**
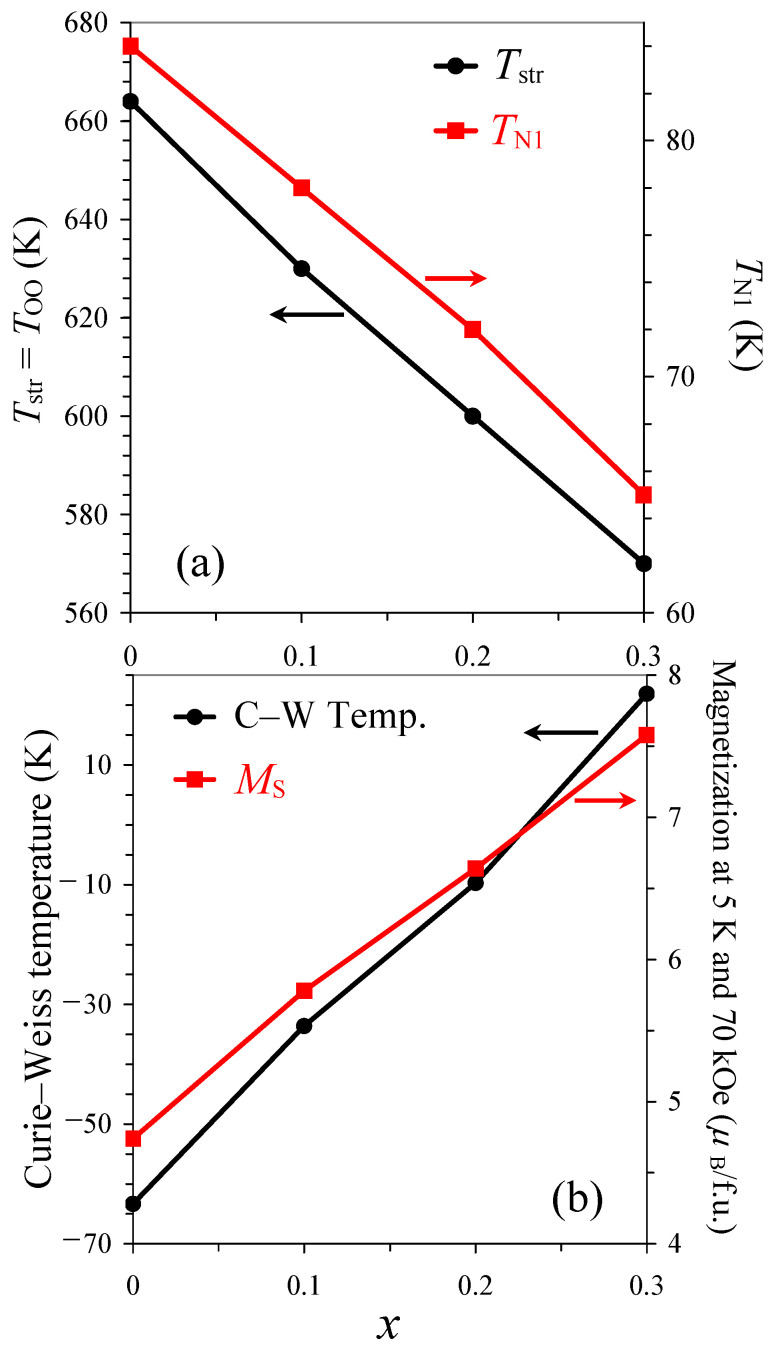
(**a**) Compositional dependence of the structural transition temperature (*T*_srt_ = *T*_OO_; black, the left-hand axis) and the first magnetic transition temperature (*T*_N1_; red, the right-hand axis) in the NdCu*_x_*Mn_7−*x*_O_12_ solid solutions. (**b**) Compositional dependence of the Curie–Weiss temperature (black, the left-hand axis) and the magnetization values (at *T* = 5 K and *H* = 70 kOe) (red, the right-hand axis).

**Figure 6 molecules-30-04561-f006:**
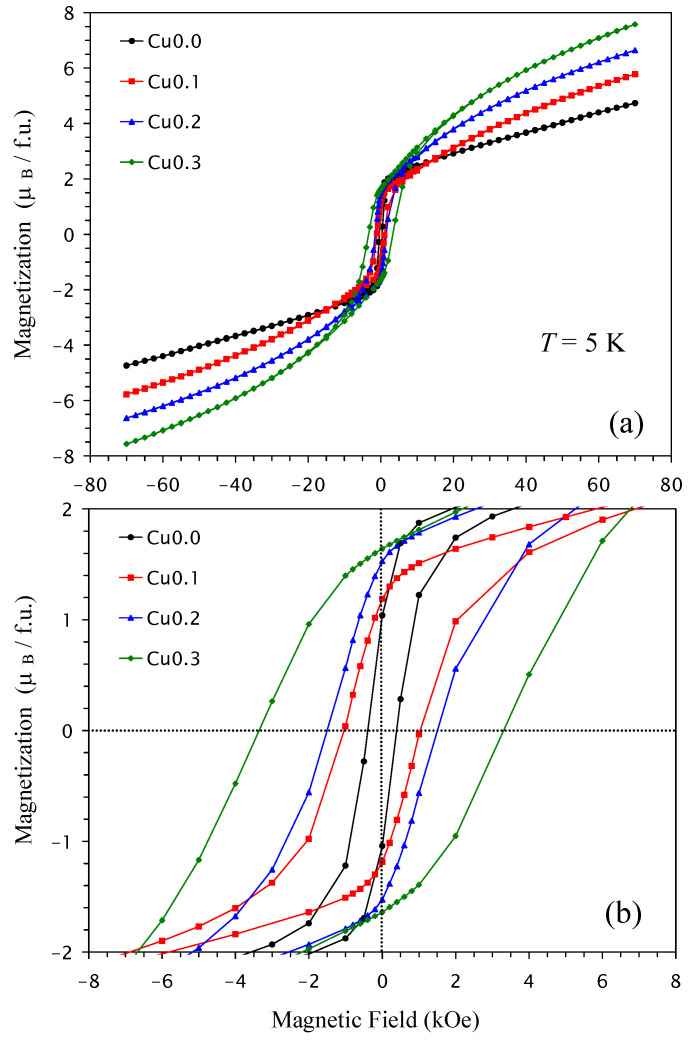
(**a**) *M* versus *H* curves of NdCu*_x_*Mn_7−*x*_O_12_ with *x* = 0 (black), 0.1 (red), 0.2 (blue), and 0.3 (green), measured at *T* = 5 K. (**b**) The same *M* versus *H* curves zoomed in on near the origin.

**Figure 7 molecules-30-04561-f007:**
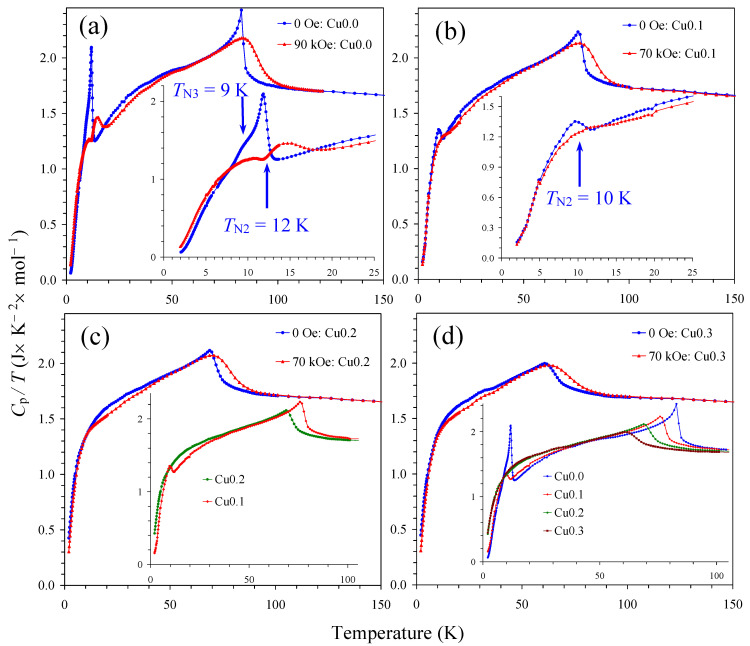
*C*_p_/*T* versus *T* curves of the NdCu*_x_*Mn_7−*x*_O_12_ solid solutions measured at *H* = 0 Oe (blue curves) and 90 kOe (or 70 kOe) (red curves) during cooling for (**a**) *x* = 0, (**b**) *x* = 0.1, (**c**) *x* = 0.2, and (**d**) *x* = 0.3. The insets on panels (**a**,**b**) show the same curves below 25 K. Arrows give the magnetic transition temperatures. The inset on panel (**c**) compares *C*_p_/*T* versus *T* curves at *H* = 0 Oe for *x* = 0.1 and *x* = 0.2. The inset on panel (**d**) compares *C*_p_/*T* versus *T* curves at *H* = 0 Oe for all samples.

**Figure 8 molecules-30-04561-f008:**
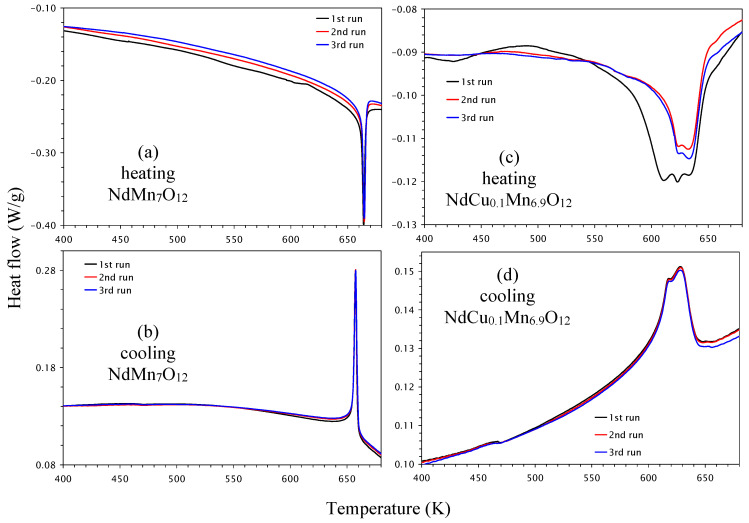
Differential scanning calorimetry (DSC) curves of (**a**,**b**) NdMn_7_O_12_ and (**c**,**d**) NdCu_0.1_Mn_6.9_O_12_ during (**a**,**c**) heating and (**b**,**d**) cooling, shown between 400 K and 680 K. Three DSC runs are shown. The anomalies near 470 K on the cooling curves are instrumental artifacts.

**Figure 9 molecules-30-04561-f009:**
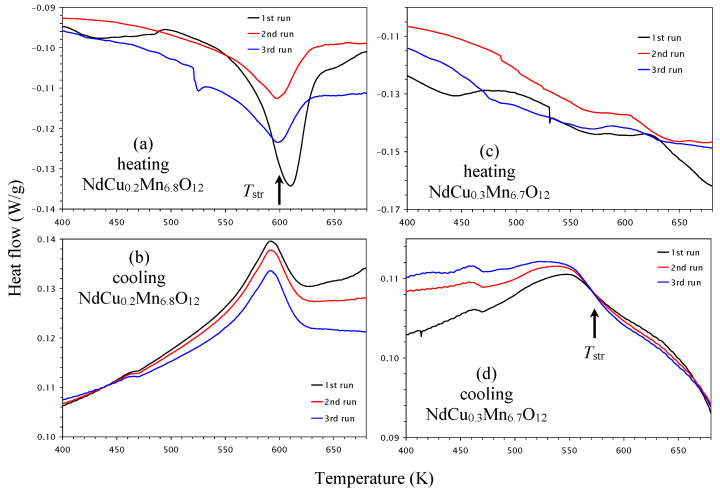
Differential scanning calorimetry (DSC) curves of (**a**,**b**) NdCu_0.2_Mn_6.8_O_12_ and (**c**,**d**) NdCu_0.3_Mn_6.7_O_12_ during (**a**,**c**) heating and (**b**,**d**) cooling, shown between 400 K and 680 K. Three DSC runs are shown. The anomalies near 470 K on the cooling curves and sharp drops on some of the heating curves are instrumental artifacts.

**Figure 10 molecules-30-04561-f010:**
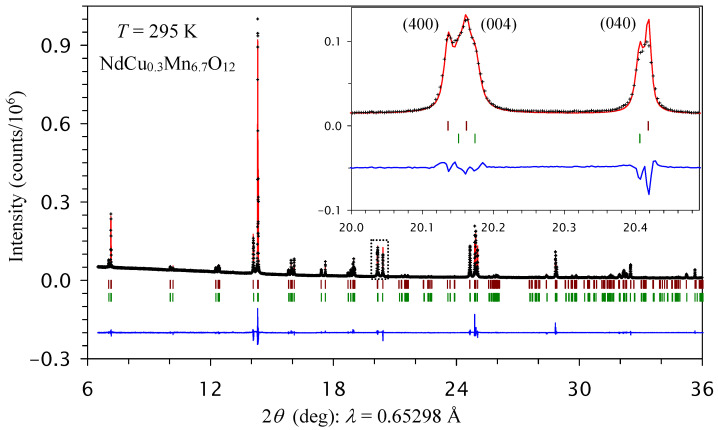
Fragments of the experimental (black crosses), calculated (red line), and difference (blue line at the bottom) room-temperature synchrotron X-ray powder diffraction patterns of NdCu_0.3_Mn_6.7_O_12_ in a 2*θ* range of 6° and 36°, analyzed by the Rietveld method. The tick marks show possible Bragg reflection positions for two *I*2/*m* phases. The inset shows a zoomed-in part in a 2*θ* range of 20.0° and 20.5° (shown by a dotted rectangle in the main panel) and emphasizes the splitting of (400), (004), and (040) reflections from phase separation.

**Figure 11 molecules-30-04561-f011:**
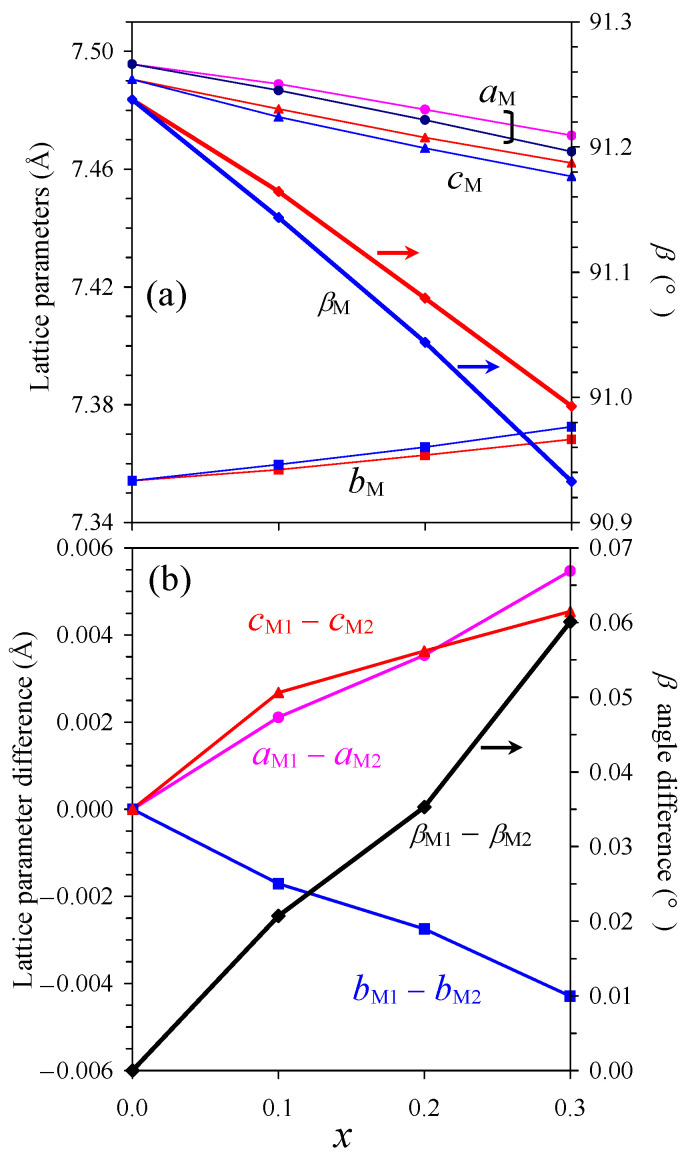
(**a**) Compositional dependence of the monoclinic (M) lattice parameters (the left-hand axis) and monoclinic *β* angle (the right-hand axis) in NdCu*_x_*Mn_7−*x*_O_12_ with *x* = 0, 0.1, 0.2, and 0.3. The lattice parameters of the first *I*2/*m* phase (M1) are shown in red and pink, and the lattice parameters of the second *I*2/*m* phase (M2) are shown in blue and navy. (**b**) Compositional dependence of the difference in the lattice parameters between phases M1 and M2.

**Table 1 molecules-30-04561-t001:** Temperatures of structural transitions (*T*_str_) and magnetic anomalies (*T*_N_) and parameters of the Curie–Weiss fits and *M* versus *H* curves at *T* = 5 K for NdCu*_x_*Mn_7−*x*_O_12_ with *x* = 0, 0.1, 0.2, and 0.3.

*x*	*T*_str_ (K)	*T*_N_ (K)	*μ*_eff_ (*μ*_B_/f.u.)	*μ*_calc_ (*μ*_B_/f.u.)	*θ* (K)	*M*_S_ (*μ*_B_/f.u.)	*M*_R_ (*μ*_B_/f.u.)	*H*_C_ (kOe/f.u.)
0	664	9, 12, 84	13.651	13.426	−63.3	4.74	1.04	~0.4
0.1	~630	10, 78	13.125	13.314	−33.6	5.78	1.19	~1.0
0.2	~600	72	12.977	13.200	−9.7	6.64	1.53	~1.5
0.3	~570	65	12.561	13.086	+21.9	7.58	1.64	~3.3

The Curie–Weiss fits were performed between 200 and 350 K using the FCC *χ*^−1^ versus *T* data at 10 kOe. *M*_S_ is the magnetization value at *T* = 5 K and *H* = 70 kOe. *M*_R_ is the remnant magnetization value at *T* = 5 K. *H*_C_ is the coercive field at *T* = 5 K. *μ*_eff_ is an experimental effective magnetic moment. *μ*_calc_ is a calculated effective magnetic moment based on the formal oxidation states. *T*_N_ values were determined from peaks on the 100 Oe and 10 kOe FCC *d*(*χT*)/*dT* versus *T* curves. *T*_str_ values were determined from peak positions on the heating (or cooling for *x* = 0.3) DSC curves. *T*_str_ corresponds to a transition to the *Im*-3 modification.

**Table 2 molecules-30-04561-t002:** Lattice parameters for NdCu*_x_*Mn_7−*x*_O_12_ with *x* = 0, 0.1, 0.2, and 0.3 at room temperature (space group *I*2/*m*), determined from high-resolution synchrotron powder X-ray diffraction. Lattice parameters for two *I*2/*m* phases (M1 and M2) are given for *x* = 0.1, 0.2, and 0.3.

*x*	Phase	*a* (Å)	*b* (Å)	*c* (Å)	*β* (°)	*V* (Å^3^)
0	M1	7.49567 (1)	7.35419 (1)	7.49049 (1)	91.2384 (2)	412.814 (1)
0.1	M1	7.48888 (2)	7.35795 (2)	7.48042 (3)	91.1644 (3)	412.107 (2)
	M2	7.48677 (3)	7.35966 (3)	7.47774 (3)	91.1437 (2)	411.942 (3)
0.2	M1	7.48026 (3)	7.36286 (2)	7.47075 (3)	91.0793 (3)	411.387 (2)
	M2	7.47672 (2)	7.36561 (2)	7.46712 (2)	91.0440 (2)	411.151 (2)
0.3	M1	7.47143 (2)	7.36828 (2)	7.46210 (2)	90.9929 (3)	410.739 (2)
	M2	7.46596 (3)	7.37257 (3)	7.45756 (3)	90.9328 (4)	410.434 (3)

## Data Availability

The raw data supporting the conclusions of this article will be made available by the author on request.

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
