# Peer review of "Phase Separation Phenomena in Lightly Cu-Doped A-Site-Ordered Quadruple Perovskite NdMn_7_O_12"

_molecules, 2025, doi:10.3390/molecules30234561_

Round 1
Reviewer 1 Report
Comments and Suggestions for Authors
Reviewer Report
In the reviewed manuscript, the authors present a comprehensive experimental investigation of lightly Cu-doped A-site-ordered quadruple perovskite NdMn₇O₁₂. The combination of HPHT synthesis, synchrotron XRD, DSC, magnetic measurements and specific-heat data results in a coherent and convincing dataset. The evidence for phase separation into two closely related monoclinic I2/m phases in the doping range x = 0.1–0.3 is solid, and the work fits well within the scope of the journal.
However, several aspects of the interpretation would benefit from further clarification or extension, especially regarding the microscopic origin of the phase separation and the connection between specific measurements and the conclusions drawn. These points are summarized below.
Comments line-by-line
- Please expand the explanation of the mechanism behind the observed phase separation. It is unclear why the effect appears only for x = 0.1–0.3 and not at higher Cu contents.
• Clarify whether Cu²⁺ can be considered fully confined to the A′ site, and comment on whether Mn³⁺/Mn⁴⁺ distribution or lattice strain could contribute to the phase separation.
• For the DSC data, specify whether the relative peak areas qualitatively correspond to the approximately 1:1 phase ratio obtained from the Rietveld refinements.
• Add a short explanation of why the two coexisting phases do not produce distinguishable magnetic anomalies in the magnetization data.
• Consider adding coercive field (Hc) values to Table 1, together with MS and MR, to provide a more complete magnetic characterization.
• In Table 1, explicitly state that μcalc refers to the calculated magnetic moment based on formal oxidation states.
• Correct minor typographical issues, for example “effected” → “affected” and “irreducibility” → “irreproducibility”.
• Some figure insets, such as in Fig. 7, have rather small axis labels; increasing the font size would improve readability.
• For Fig. 10, adding a scale bar or specifying the width of the zoomed 2θ range directly in the caption would help with interpretation.
• In the Introduction, consider adding one sentence explaining why Cu²⁺ prefers square-planar coordination at the A′ site (Jahn–Teller stabilization).
Author Response
Reviewer 1.
- Please expand the explanation of the mechanism behind the observed phase separation. It is unclear why the effect appears only for x = 0.1–0.3 and not at higher Cu contents.
Our reply.
We should say that at the moment we do not have a clear and unambiguous explanation of the mechanism behind the observed phase separation. Therefore, in the revised manuscript, we made the following comments: “The real reason of phase separation in the 0.1–0.3 samples is not clear and will require additional studies in the future.” “We also note that phase separation was reported in cubic CdCu3Mn4O12 [45], prepared under high-pressure conditions; however, in this case, the formation of CuO impurity (and compositional shifts) could have a contribution to the effect. Phase separation was also observed in some of cubic CaCu3Ti4O12 samples [68], prepared at ambient pressure.”
- Clarify whether Cu²⁺ can be considered fully confined to the A′ site, and comment on whether Mn³⁺/Mn⁴⁺ distribution or lattice strain could contribute to the phase separation.
Our reply.
The first part of this comment is related to comment 10. In the revised manuscript, we gave the following clarifications: “We emphasize that previous structural studies with neutron diffraction [44, 51, 53, 56, 58, 59, 61] showed that Cu2+ cations are always localized at the square-planar A' site due to the strong Jahn–Teller effect of Cu2+.” “However, Mn³⁺/Mn⁴⁺ distribution and ratio (which depends on the oxygen content and Cu2+ doping levels) or lattice strain could also contribute to the phase separation.”
- For the DSC data, specify whether the relative peak areas qualitatively correspond to the approximately 1:1 phase ratio obtained from the Rietveld refinements.
Our reply.
We thank the reviewer for this suggestion. The DSC results (intensities of the peaks) are indeed additional evidence which supports the results of synchrotron studies. Therefore, in the revised manuscript, we added the following clarification: “The intensities of the two DSC peaks in the x = 0.1 sample (on cooling curves and on the second and third heating curves) were comparable with each other in agreement with an approximate 1:1 ratio of the two monoclinic phases found by synchrotron PXRD.”
- Add a short explanation of why the two coexisting phases do not produce distinguishable magnetic anomalies in the magnetization data.
Our reply.
The following explanation was given in the revised manuscript: “….probably because magnetic transition temperatures of the two monoclinic phases are too close to each other, and all magnetic and specific heat anomalies from two phases severely overlap.”
- Consider adding coercive field (Hc) values to Table 1, together with MS and MR, to provide a more complete magnetic characterization.
Our reply.
As the reviewer suggested, we added a new column with the coercive fields into the revised Table 1.
- In Table 1, explicitly state that μcalc refers to the calculated magnetic moment based on formal oxidation states.
Our reply.
In the revised Table 1, it was stated as the reviewer suggested.
- Correct minor typographical issues, for example “effected” → “affected” and “irreducibility” → “irreproducibility”.
Our reply.
We thank the reviewer for noticing these misprints. They were corrected in the revised manuscript.
- Some figure insets, such as in Fig. 7, have rather small axis labels; increasing the font size would improve readability.
Our reply.
Figure 7 was revised as suggested by the reviewer.
- For Fig. 10, adding a scale bar or specifying the width of the zoomed 2θ range directly in the caption would help with interpretation.
Our reply.
Figure 10 was revised as suggested by the reviewer.
- In the Introduction, consider adding one sentence explaining why Cu²⁺ prefers square-planar coordination at the A′ site (Jahn–Teller stabilization).
Our reply.
In the revised introduction of the manuscript, we gave the following clarification: “We emphasize that previous structural studies with neutron diffraction [44, 51, 53, 56, 58, 59, 61] showed that Cu2+ cations are always localized at the square-planar A' site due to the strong Jahn–Teller effect of Cu2+.”
Reviewer 2 Report
Comments and Suggestions for Authors
The manuscript is interesting and could be published with the following additions:
The separation of two phases having the same space group, similar but not identical composition and slightly different lattice constants is a very interesting problem. You explained this by saying that they have slightly different oxygen concentrations.
In figure 10 , the X-ray powder diffraction pattern was fitted using the Rietveld method or what method did you use to compare the calculated and experimental powder diffraction pattern?
The calculation program used for both indexing and Rietveld refinement should be specified if it was done. Indexing could be done with the X-Cell computing program from the Material Studio software package which is a reliable indexing program.
If you have the opportunity and the time available, you could try a Rietveld refinement to determine the structural model for the two phases and determine the positions of the oxygen atoms in the two phases.
Author Response
Reviewer 2.
- In figure 10 , the X-ray powder diffraction pattern was fitted using the Rietveld method or what method did you use to compare the calculated and experimental powder diffraction pattern?
Our reply.
In the revised caption of Figure 10, we specified that X-ray powder diffraction pattern was fitted using the Rietveld method.
- The calculation program used for both indexing and Rietveld refinement should be specified if it was done. Indexing could be done with the X-Cell computing program from the Material Studio software package which is a reliable indexing program.
Our reply.
The Rietveld refinement program was mentioned in the materials and methods part as “The Rietveld analysis of all XRPD data and indexing were performed using the RIETAN-2000 program [70].”
- If you have the opportunity and the time available, you could try a Rietveld refinement to determine the structural model for the two phases and determine the positions of the oxygen atoms in the two phases.
Our reply.
We thank the reviewer for this suggestion. The refined structural parameters are reported in the supporting information file of the revised manuscript.
Reviewer 3 Report
Comments and Suggestions for Authors
The submission reports a typical structural study of a specific material: lightly Cu-doped A-site-ordered quadruple perovskite NdMn_7O_12. A dependence on the Cu doping level has been observed, and a variation in the number of structural transitions has been noted with increasing doping. The experimental study has utilized observations of magnetoresistance properties, ferrimagnetism, and additional structural modulations which produce electric-dipole helicoidal textures via differential scanning calorimetry measurements and high-resolution synchrotron powder X-ray diffraction studies. These are standard methods to analyze structural transitions in materials. No application significance of the study and material has been indicated. The paper is well written, and the results are clearly reported and illustrated. It can be published as it contributes to the study of this particular material, with minor revision, however.
In line 321, it should be ‘synthesized’ not ‘synthesize’ – the paper needs proofreading.
Some comments about applications of the material under study would be beneficial.
Author Response
Reviewer 3.
- In line 321, it should be ‘synthesized’ not ‘synthesize’ – the paper needs proofreading
Our reply.
We thank the reviewer for noticing this misprint. It was corrected in the revised manuscript in addition to other minor misprints.
- Some comments about applications of the material under study would be beneficial.
Our reply.
One possible application was mentioned in our manuscript as “….produces ferrimagnetic (FiM) properties with high magnetic transition temperatures above RT.” In the revised manuscript, we also added another possible application as “Such A2+Mn7O12 and A3+Mn7O12 manganites also show good catalytic properties [10].”